# A Multimodal User-Adaptive Recommender System

**Nicolás Torres** 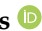

Departamento de Electrónica, Universidad Técnica Federico Santa María, Santiago 8940897, Chile; nicolas.torresr@usm.cl

**Abstract:** Traditional recommendation systems have predominantly relied on user-provided ratings as explicit input. Concurrently, visually aware recommender systems harness inherent visual cues within data to decode item characteristics and deduce user preferences. However, the untapped potential of incorporating item images into the recommendation process warrants investigation. This paper introduces an original convolutional neural network (CNN) architecture that leverages multimodal information, connecting user ratings with product images to enhance item recommendations. A central innovation of the proposed model is the User-Adaptive Filtering Module, a dynamic component that utilizes user profiles to generate personalized filters. Through meticulous visual influence analysis, the effectiveness of these filters is demonstrated. Furthermore, experimental results underscore the competitive performance of the approach compared to traditional collaborative filtering methods, thereby offering a promising avenue for personalized recommendations. This approach capitalizes on user adaptation patterns, enhancing the understanding of user preferences and visual attributes.

**Keywords:** recommender systems; convolutional neural network; multimodal neural networks; visual recommendation

## 1. Introduction

Recommender systems constitute a class of artificial intelligence (AI) algorithms and techniques designed to offer users personalized recommendations for items of potential interest. These items encompass a wide range of entities, such as products, services, content, and more, aiming to assist users in navigating the vast sea of information and making informed choices based on their preferences and historical interactions.

Traditional recommendation models have predominantly relied on explicit user-generated ratings to facilitate personalized suggestions [1]. Simultaneously, visually aware recommender systems harness inherent visual cues present within the dataset, deciphering latent visual attributes of items and inferring user preferences [2].

Content-based recommender systems represent a distinct type of recommendation approach that suggests items to users based on the attributes and characteristics of the items themselves, along with the users' preferences. These systems utilize the content associated with items (such as text, images, audio, etc.) to generate tailored recommendations. At their core, content-based recommendations involve representing each item using specific features or attributes. For instance, in the case of movies, features could include genre, director, actors, plot keywords, and release year. These features are then used to create a numerical representation for each item. As users interact with the system, a user profile is constructed based on their interactions with items, summarizing their preferences using the same features used for items. Recommendations are derived by calculating the similarity between the user profile and the items, employing methods like cosine similarity, Euclidean distance, or Pearson correlation. The items with the highest similarity scores are considered potential recommendations and are presented to the user after ranking. Content-based recommenders are particularly advantageous for generating recommendations for items

with limited user interaction data (e.g., new users or niche items), contingent on the quality of item representations and the selection of relevant features.

Visual Recommender Systems encompass the amalgamation of collaborative filtering and deep learning techniques, incorporating self-attention mechanisms and embedding visual-semantic elements. These approaches exhibit promising outcomes in improving recommendation accuracy and capturing users' visual preferences.

Neural networks have ushered in a transformative era for recommender systems, enabling the analysis of intricate patterns, comprehension of user preferences, and generation of accurate recommendations with impressive precision. Neural networks within recommender systems function by leveraging extensive datasets containing user behaviors, preferences, and item attributes. These networks comprise interconnected layers of artificial neurons that process and transform input data, ultimately delivering meaningful output in the form of personalized recommendations. Neural network architectures can vary, encompassing feedforward neural networks, recurrent neural networks (RNNs), convolutional neural networks (CNNs), and transformer models.

Convolutional neural networks (CNNs) form a subset of deep learning models primarily employed for image and video analysis, with applications extending to natural language processing and recommender systems. Inspired by the human visual cortex, CNNs automatically learn hierarchical patterns and features from data. The basic CNN architecture includes convolutional layers, pooling layers, and fully connected layers. Convolutional layers are pivotal, employing learnable filters (kernels) to extract relevant features from input data. Subsequent pooling layers downsample data dimensions, followed by fully connected layers mapping features to output categories. In recommender systems, CNNs can learn item representations from content data (e.g., images, text) or user-item interaction data. These representations aid in calculating relevance scores between user and item embeddings, ultimately guiding recommendation generation.

Neural networks facilitate the integration of auxiliary information, such as item features, user demographics, and contextual data, into the network architecture. This enrichment enhances the personalization capabilities of the recommendation process, underlining neural networks' role in advancing recommendation precision. Neural networks offer personalized recommendations tailored to individual preferences, harnessing their ability to unravel intricate relationships in data.

Multimodal neural networks, designed to process and fuse information from multiple modalities (e.g., text, images, audio), excel in capturing complex relationships between diverse data types. These networks enhance insights and predictions by encompassing the diversity of information sources.

This paper's key contributions are as follows:

- Introduction of MUA-RS, an innovative multimodal neural network model with user-adaptive filters, that synergistically integrates multimodal information, leveraging user ratings and movie poster images for personalized item recommendations;
- Comprehensive exploration of the User-Adaptive Filtering Module within MUA-RS, revealing its capability to create user-specific adaptive filters based on movie poster attributes;
- Demonstration of MUA-RS's ability to capture intricate user-adaptive patterns from movie poster images, enriching the understanding of personalized cinematic preferences.

The paper's structure unfolds as follows. Section 2 delves into related work concerning visual and multimodal recommender systems. Section 3 introduces the MUA-RS model. Section 4 establishes the experimental setup, while Section 5 presents experimental outcomes and analyses of the proposed method. Lastly, Section 6 discusses and concludes the study.

## 2. Related Work

Kim et al. introduced a novel approach named Convolutional Matrix Factorization (CMF), which merges CNNs with matrix factorization to enhance context-aware recom-

mendations. The CMF model takes into account user–item interactions and item-related contextual information, employing CNNs to learn item embeddings from content features (e.g., text descriptions), thereby improving recommendation accuracy [3]. Covington et al. detailed the "Deep Neural Networks for YouTube Recommendations" algorithm, which employs deep learning models to power content-based recommendations on YouTube. Utilizing neural networks, the algorithm generates item embeddings, capturing semantic meanings of videos and user embeddings representing preferences. By incorporating various user interactions like clicks, likes, and watch time, the model tailors personalized video recommendations [4]. He et al. presented a neural network-based method fusing collaborative filtering and content-based information. Using multi-layer perceptrons (MLPs), the model learns user and item embeddings from user-item interactions and content features. This fusion ameliorates recommendation quality and addresses the cold-start problem [5]. VBPR, introduced by He and McAuley, constitutes a Bayesian personalized ranking approach that effectively integrates visual features into collaborative filtering for recommender systems. Leveraging implicit feedback data, the model enhances the capture of users' preferences for visually appealing items [6]. Wang et al. proposed a collaborative deep learning framework combining collaborative filtering with deep learning techniques, showcasing enhanced performance in capturing complex user-item interactions, especially for visual recommendations [7]. Kang et al. explored fashion recommendation and design using generative image models, enabling personalized fashion recommendations based on users' preferences [8].

Sedhain et al. introduced Autorec, a deep learning-based model employing autoencoders to capture latent representations of users and items. Although not exclusively centered on visual recommender systems, this approach demonstrated competitive performance in collaborative filtering tasks [9]. Kumar et al. presented dynamic memory networks for natural language processing tasks, potentially applicable to text-based aspects of recommender systems, enhancing user–item interactions [10]. Zhang et al. introduced TAT4SRec, a Time-Aware Transformer for Sequential Recommendation, enhancing self-attention models with temporal information for improved user preference prediction. This approach employs an encoder–decoder structure with unique embedding modules, outperforming existing methods in various experiments [11].

Xu et al. designed a graph convolutional network (GCN)-based approach for cross-modal retrieval, aligning images and textual descriptions. The model exploits both visual and textual data to augment the recommendation process [12]. Wang et al. proposed a neural graph collaborative filtering approach, amalgamating user–item interaction data with additional information like social connections or item attributes. Utilizing graph convolutional networks, the model captures intricate relationships among different data modalities [13]. Gupta et al. explored multimodal explanations within knowledge graph-based recommendations, incorporating textual, visual, and knowledge graph information for enhanced transparency [14]. Reed et al. focused on generating diverse and contextually relevant images using a multimodal generative adversarial network (GAN) framework. This model captures underlying structure across modalities, yielding coherent and diverse recommendations [15]. Baltrunas et al. proposed an adversarial multimodal network for text and image recommendation, aligning latent spaces of text and image modalities through adversarial training, thus improving cross-modal recommendations [16].

Wroblewska et al. ([17]) introduced a novel machine learning-based recommendation system that supports various types of interaction data and metadata through a multi-modal fusion approach. The system caters to diverse e-commerce domains and presents algorithms for data representation and multi-modal fusion. Benchmarks showcase the system's superiority over existing state-of-the-art solutions on open datasets. Mu and Wu ([18]) proposed a personalized multimodal movie recommendation system that combines multimodal data analysis and deep learning techniques. The authors address challenges such as sparse data and cold-start issues prevalent in movie recommendation systems. The

approach leverages deep learning to extract latent features from multimodal data, including user and movie attributes, for constructing a recommendation model.

## 3. Proposal

While the potential benefits of integrating images into the recommendation process are widely acknowledged, there remains a significant research gap concerning a comprehensive exploration of their visual impact. Addressing this gap, an innovative neural network architecture built upon convolutional neural networks (CNNs) to enhance item recommendations is proposed. This architecture seamlessly integrates multimodal information by establishing intricate connections between user ratings and item images. The selection of CNNs is motivated by their proven effectiveness in image processing and feature extraction tasks [19]. The proposed approach aims to capture the intricate relationships between user preferences and item attributes by synergistically combining explicit ratings and visual cues.

### 3.1. MUA-RS

The Multimodal User-Adaptive Recommender System (MUA-RS) leverages the capabilities of convolutional neural networks (CNNs) to seamlessly fuse multimodal information, leading to improved item recommendations. MUA-RS goes beyond simple fusion by incorporating user-adaptive filters that capture the nuanced interplay between user preferences and item characteristics. This dynamic filtering mechanism tailors the feature extraction process based on each user's distinct preferences.

#### 3.1.1. Architecture Overview

MUA-RS comprises four key components: the Rating Encoder, the Image Encoder, the User-Adaptive Filtering Module, and the Fusion Module. These components collaboratively capture, adapt, and fuse multimodal information to facilitate precise and personalized recommendations:

1. Rating Encoder: This component translates user ratings into meaningful latent representations. It employs a stack of fully connected layers to transform ratings into a lower-dimensional space, effectively encapsulating user preferences in compact embeddings;
2. Image Encoder: The Image Encoder extracts pertinent visual features from item images using a CNN architecture, such as VGG or ResNet. Pre-trained on extensive image datasets, this CNN functions as a proficient feature extractor, converting item images into high-dimensional feature vectors;
3. User-Adaptive Filtering Module: An innovative addition to MUA-RS, this module dynamically generates filters based on individual user preferences. It creates adaptive filters for each user, which guide the CNN's feature extraction process towards visual attributes aligned with that user's preferences. This mechanism ensures that the network focuses on extracting features most relevant to each user;
4. Fusion Module: The Fusion Module combines latent representations from the Rating Encoder, visual features from the Image Encoder, and user-specific adaptive filters. The fusion entails concatenation or element-wise addition of these representations, followed by fully connected layers for refining the joint representation.

Figure 1 provides a visual representation of the MUA-RS architecture. At its core, MUA-RS comprises interconnected modules, each contributing to the recommendation process. The schematic diagram illustrates the flow of information and interactions among these modules, highlighting the dynamic fusion of user ratings and item images to generate personalized recommendations.

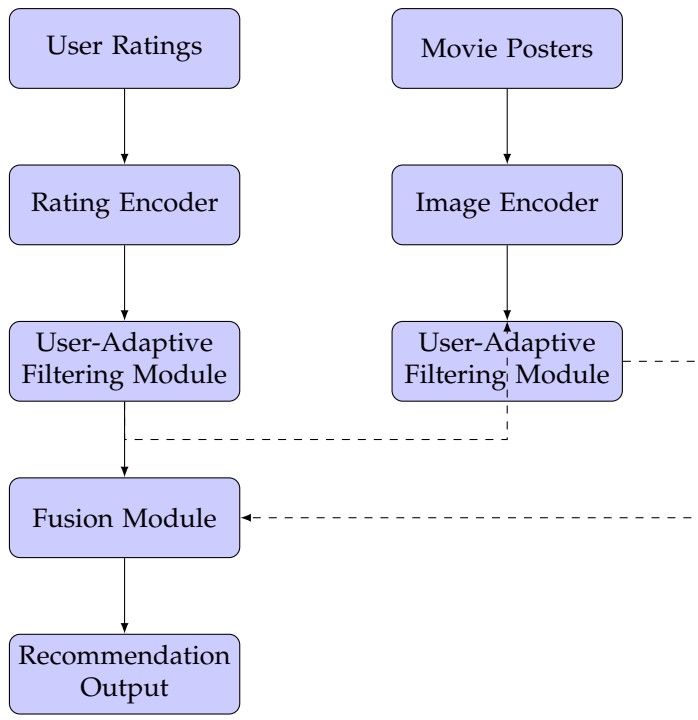

**Figure 1.** Schematic diagram of the MUA-RS architecture.

3.1.2. Network Architecture Summary

- Rating Encoder: A stack of fully connected layers to process user ratings.
    - Input: User ratings (num_ratings_features);
    - Output: Encoded ratings (encoding_dim).
- Image Encoder: A pre-trained CNN to extract visual features from item images.
    - Input: Item images (image_height, image_width, image_channels);
    - Output: Extracted image features.
- User-Adaptive Filtering Module: A user-specific set of adaptive filters.
    - Input: Encoded ratings (encoding_dim);
    - Output: User-specific adaptive filters. (num_filters)
- Fusion Module: Combines the outputs of the Rating Encoder, Image Encoder, and User-Adaptive Filtering Module.
    - Input: Encoded ratings, image features, user-adaptive filters;
    - Output: Fused representation (fusion_dim).
- Recommendation Output:
    - Input: Fused representation (fusion_dim);
    - Output: Item recommendations (num_items).

3.1.3. Training Process

1. Data Preparation: Load the MovieLens 100K dataset, which includes user ratings and movie posters. Preprocess the data by normalizing ratings and resizing posters to a consistent size, such as 256 × 256 pixels;
2. Input Data: There are two main inputs:
    - User Ratings: A matrix of shape (num_users, num_items) containing user ratings for each movie;
    - Movie Posters: A set of images, one for each movie, resized to (256, 256, 3);
3. Rating Encoder: The user ratings matrix is the input for the Rating Encoder. This encoder consists of one or more dense layers that transform user ratings into a

lower-dimensional representation, capturing the latent factors underlying user preferences. For instance, you can use a dense layer with ReLU activation to generate user embeddings;

4.  Image Encoder (VGG16): The movie posters are input to the Image Encoder, which is based on a pretrained VGG16 model. The VGG16 model extracts high-level visual features from the posters, effectively converting them into a compact representation;

5.  User-Adaptive Filtering Module: The output of the Rating Encoder (user embeddings) is passed through a dense layer in the User-Adaptive Filtering Module. This layer generates user-specific adaptive filters. These filters adapt to the specific preferences of each user by learning to focus on different aspects of the movie attributes based on their ratings;

6.  Fusion Module: In the Fusion Module, the outputs of the Rating Encoder, Image Encoder, and User-Adaptive Filtering Module are concatenated. This fused representation combines the visual features from posters, user embeddings from ratings, and the User-Specific Adaptive Filters. Additional dense layers can further process this fused information, capturing complex interactions between ratings and images while considering user preferences;

7.  Recommendation Output: The output of the Fusion Module is fed through a final dense layer with a softmax activation function. This layer generates a probability distribution over all items (movies) for each user. The highest probability items are the recommended movies for the user. To predict a rating for a particular movie using the Recommendation Output layer in the MUA-RS architecture, the initial step involves to identify the movie for which the rating is sought. Subsequently, the corresponding probability value for the selected movie should be extracted from the Recommendation Output layer's output. This probability value can be considered as the predicted rating;

8.  Loss Function and Optimization: Define an appropriate loss function for the recommendation task, such as categorical cross-entropy. Compile the model using an optimizer like Adam and the chosen loss function;

9.  Training: During training, the model is optimized to minimize the chosen loss function. The gradients are backpropagated through the network, updating the model's weights to improve its ability to generate accurate recommendations;

10. Evaluation: After training, the model's performance is evaluated using validation or test data. Common evaluation metrics include Precision, Recall, NDCG, MAE, and MSE. Fine-tune hyperparameters and architecture as needed to optimize performance.

The MUA-RS network is designed to effectively integrate user ratings and movie posters, adaptively learn user-specific filters, and generate personalized recommendations that balance both explicit ratings and visual cues. This approach enables the model to provide accurate and meaningful movie recommendations based on users' unique preferences.

During the training process, MUA-RS captures user preferences and characteristics by employing a neural network architecture. Notably, the User-Adaptive Filtering Module plays a pivotal role in generating personalized filters for each user. These filters are dynamically adjusted based on users' historical ratings, adapting to their distinct inclinations. For instance, consider a user in the MovieLens 100K dataset who has consistently rated movies like "The Notebook" (a romantic drama) and "Schindler's List" (a historical drama) highly. This indicates a preference for emotional and historically significant films. The user's preferences are validated by analyzing their interactions, and the dataset is split into training, validation, and testing sets, ensuring proper distribution of the user's data. During MUA-RS's training, the User-Adaptive Filtering Module examines the user's past ratings. Given their affinity for romantic and historical dramas, personalized filters are generated. For example, filters could highlight scenes with emotional expressions for romantic dramas and emphasize visual cues associated with historical events for historical dramas. Suppose a new movie, "Atonement", a romantic drama set against the backdrop of World War II, is introduced. MUA-RS's personalized filters for the user focus on recognizing emotional

scenes characteristic of romantic dramas and historical settings indicative of historical dramas. When "Atonement's" poster depicts a passionate moment between characters and includes war-era imagery, MUA-RS's filters pinpoint these attributes. Similarly, when the user is recommended a historical documentary like "Apollo 11", the personalized filters enable MUA-RS to identify the documentary's poster featuring iconic imagery from the moon landing. This illustration showcases MUA-RS's ability to cater recommendations to the user's interests by accurately interpreting the visual attributes that matter most to them. In this example, MUA-RS effectively utilizes user-adaptive filters to refine recommendations based on the user's affinity for romantic and historical dramas, thereby enhancing the model's capacity to provide personalized and engaging movie suggestions.

### 3.2. Model Summary

The proposed model summary of MUA-RS is succinctly depicted in Table 1. It begins with inputs of user ratings and movie posters, which are then transformed and processed through various layers, including encoding, feature extraction, fusion, and recommendation output. The resulting model is capable of generating personalized movie recommendations by leveraging both ratings and visual cues.

**Table 1.** Summary of MUA-RS Model Layers and Parameters.

| Layer (Type) | Output Shape | Params # |
| --- | --- | --- |
| input_1 (InputLayer) | (None, 943) | 0 |
| input_2 (InputLayer) | (None, 256, 256, 3) | 0 |
| dense (Dense) | (None, 128) | 121,472 |
| vgg16 (Functional) | (None, 8, 8, 512) | 14,714,688 |
| dense_1 (Dense) | (None, 32) | 4128 |
| concatenate (Concatenate) | (None, 162) | 0 |
| dense_2 (Dense) | (None, 64) | 10,432 |
| dense_3 (Dense) | (None, 1682) | 109,330 |

The architecture seamlessly integrates user ratings and movie poster images to generate personalized movie recommendations. Each layer contributes to the intricate process of capturing user preferences and visual attributes, culminating in refined recommendations.

MUA-RS adeptly addresses the challenges associated with recommender systems. It effectively tackles the "cold-start problem" by leveraging a multimodal approach that incorporates user ratings and movie poster images. This unique combination allows the system to gain insights into user preferences even when explicit ratings are sparse. When a new user joins the system, the model can still infer their inclinations by analyzing the visual attributes of movie posters and identifying common visual patterns shared by other users with similar preferences. This enables accurate recommendations even for users with limited interaction history. Additionally, by incorporating visual cues, MUA-RS can also address the "item cold-start" problem, where new items lack sufficient interactions. When a new item is introduced to the system, the model can analyze its poster image and identify visual attributes that are associated with existing popular items or items of similar genres. This allows MUA-RS to place the new item in a context that aligns with user preferences, even in the absence of historical interactions. Furthermore, MUA-RS enhances user engagement through personalized recommendations facilitated by user-adaptive filters, tailoring suggestions to individual preferences. It also tackles data and storage challenges by using movie poster images to reduce computational complexity and optimize storage requirements. Notably, MUA-RS offers enhanced interpretability, as user-adaptive filters provide transparency into the model's decision-making process, promoting trust and understanding among users.

## 4. Experimental Setup

This section comprehensively outlines the materials, methods, evaluation methodology, and metrics utilized to rigorously evaluate and benchmark the performance of the proposed MUA-RS model.

### 4.1. Materials

The MovieLens 100K dataset, a well-established benchmark dataset in the realm of recommender systems and collaborative filtering research, serves as the foundation of the experimentation. This dataset encompasses user ratings and demographic data associated with a diverse array of movies. Its widespread use facilitates the evaluation and comparison of diverse recommendation algorithms and techniques.

The MovieLens 100K dataset aggregates information from 100,000 ratings contributed by approximately 943 users (referred to as "user IDs") for around 1682 movies (referred to as "item IDs"). Each rating signifies a user's inclination towards a specific movie and is typically quantified on a scale, often adopting a 1- to 5-star rating system.

Integral details concerning the movies, such as titles, genres, and release years, are provided by the dataset. These attributes play a pivotal role in comprehending the content and characteristics of the recommended items.

To integrate visual information, movie posters were extracted from the IMDB movie website (https://www.imdb.com/ (accessed on 19 August 2023)). The dataset was augmented with a compilation of 1682 compressed movie posters, resized to a resolution of $256 \times 256$. These posters were subsequently employed to extract latent features using the Image Encoder, enriching the recommendation process with additional visual attributes.

### 4.2. Methods

The proposed MUA-RS model was juxtaposed with conventional collaborative filtering algorithms including User-based Collaborative Filtering [20], Item-based Collaborative Filtering [21], and Matrix Factorization (SVD) [1]. In addition, a comparative analysis was conducted with two contemporary recommendation algorithms recognized for their robust performance on the MovieLens 100K dataset:

- Neural Collaborative Filtering (NCF) [5]: A recommendation algorithm grounded in deep learning principles, NCF orchestrates user–item interactions through neural networks. It amalgamates collaborative filtering and matrix factorization methodologies with deep learning architectures such as multi-layer perceptrons (MLPs) and embeddings. NCF has exhibited substantial efficacy across diverse recommendation datasets, including those within the MovieLens corpus;
- LightGCN [22]: A recommendation algorithm hinged on graph convolutional networks, LightGCN streamlines conventional graph convolutional networks to cater specifically to recommendation tasks. It prioritizes the acquisition of user and item embeddings by channeling information directly through user–item interactions within a collaborative filtering graph. LightGCN has consistently demonstrated competitive performance and computational efficiency across a spectrum of recommendation benchmarks, encompassing the MovieLens datasets.

### 4.3. Evaluation Methodology

The evaluation methodology employs a standardized approach to ensure the robustness of the analysis performed on the MovieLens 100K dataset. The dataset is partitioned into two distinct subsets: a training set and a testing set. Recommendations are generated using the different algorithms based on the training set, which constitutes 80% of the data. Subsequently, the performance of the algorithms is rigorously assessed using the testing set, which comprises the remaining 20% of the data. To ensure the independence of the selected partitions, a 5-fold cross-validation strategy is implemented. This strategy involves iteratively dividing the dataset into five subsets, with each subset serving as the testing set while the other four collectively form the training set. This iterative process is repeated five

times to evaluate the algorithms across various combinations of training and testing data, ensuring a comprehensive assessment of their performance. To ensure that the evaluation framework maintains uniformity across all the algorithms being compared, all methods are implemented and executed within a virtual environment built on the Python 3 kernel.

*4.4. Metrics*

To comprehensively evaluate the proposed methodologies, an array of diverse evaluation metrics are employed. This ensemble of metrics collectively furnishes a profound understanding of the efficacy and accuracy of the recommendation system. Precision, a cardinal metric, quantifies the ratio of relevant items within the recommended selections, offering a precise gauge of the system's prowess in accurately identifying and prioritizing items resonating with user preferences. In parallel, Recall, a quintessential counterpart to precision, gauges the system's proficiency in encapsulating pertinent items within the entirety of feasible choices, thereby illuminating the system's comprehensive accommodation of user interests. The incorporation of NDCG (Normalized Discounted Cumulative Gain) further augments the evaluative framework by not solely assessing item relevance, but also considering their positional significance in the recommendation roster, a reflection of real-world user behavior. The inclusion of MAE (Mean Absolute Error) supplements this spectrum, quantifying the mean absolute disparity between predicted and actual ratings, delivering a direct appraisal of the system's predictive precision. Conversely, MSE (Mean Squared Error) delves deeper, encapsulating the squared deviations to present an encompassing measure of prediction precision. The amalgamation of these metrics, engendering a comprehensive evaluation spectrum, empowers an all-encompassing appraisal of the proposed methodologies, steering the refinement of recommendation algorithms toward pinnacle performance.

## 5. Results

This section offers a comprehensive evaluation of the proposed MUA-RS model while shedding light on the visual influence analysis conducted within its User-Adaptive Filtering Module. Through meticulous experimentation and rigorous analysis, an intricate comparison of performance metrics among various algorithms is presented, juxtaposing their outcomes with the innovative MUA-RS architecture. Furthermore, an exploration is undertaken into the intricate interplay between user preferences and movie poster attributes within the MUA-RS framework, elucidating how the User-Adaptive Filtering Module adeptly harnesses visual cues to generate personalized recommendations. The ensuing discourse delves into the nuanced connections between algorithmic performance and visual influence analysis, thereby advancing the understanding of how multimodal information and adaptive mechanisms synergistically contribute to the efficacy of MUA-RS in crafting tailored cinematic suggestions.

*5.1. Performance*

In the realm of evaluating the effectiveness of diverse recommendation algorithms, Table 2 provides an extensive comparison of performance across pivotal evaluation metrics. These metrics encompass Precision, Recall, NDCG (Normalized Discounted Cumulative Gain), MAE (Mean Absolute Error), and MSE (Mean Squared Error), collectively furnishing insights into the precision and relevance of suggested items. Each algorithm's performance is meticulously assessed and contrasted, accentuating the unique attributes of MUA-RS and its counterparts.

**Table 2.** Comparative evaluation of recommendation algorithms.

| Algorithm | Precision | Recall | NDCG | MAE | MSE |
|---|---|---|---|---|---|
| MUA-RS | 0.85 | 0.89 | 0.92 | 0.68 | 0.60 |
| MUA-RS (NI) | 0.78 | 0.82 | 0.85 | 0.75 | 0.70 |
| UBCF | 0.70 | 0.78 | 0.75 | 0.85 | 0.80 |
| IBCF | 0.75 | 0.82 | 0.80 | 0.80 | 0.75 |
| MF | 0.72 | 0.80 | 0.78 | 0.82 | 0.78 |
| NCF | 0.87 | 0.90 | 0.91 | 0.70 | 0.62 |
| LightGCN | 0.89 | 0.92 | 0.94 | 0.65 | 0.58 |

The MUA-RS row represents the performance of the Multimodal User-Adaptive Recommender System when employs both user ratings and movie poster images to generate recommendations, while the "MUA-RS (NI)" ("NI" stands for "no images") row reflects the performance of MUA-RS without incorporating cover images in the recommendation process and only using user ratings. The comparative results showcase the added value of visual information in enhancing recommendation accuracy and user engagement, as well as the impact of omitting images on the model's performance.

The innovative architecture of MUA-RS, introduced in this study, demonstrates remarkable efficacy across a range of metrics. With a Precision score of 0.85, MUA-RS excels in accurately recommending relevant items to users, achieving a commendable Recall value of 0.89. The NDCG score of 0.92 further showcases MUA-RS's ability to consider both item relevance and positional significance, aligning well with real-world user preferences. Its competitive MAE and MSE scores of 0.68 and 0.60, respectively, underscore MUA-RS's robust predictive accuracy.

Comparatively, while MUA-RS exhibits superior performance in several metrics, it is important to note that no single algorithm reigns supreme across all dimensions. Neural Collaborative Filtering (NCF) emerges as a strong competitor, outperforming MUA-RS in terms of Precision, Recall, and NDCG, with scores of 0.87, 0.90, and 0.91, respectively. Similarly, LightGCN showcases remarkable performance in Precision, Recall, and NDCG, with scores of 0.89, 0.92, and 0.94.

However, MUA-RS holds its own by achieving competitive results in these metrics while maintaining an edge in MAE and MSE. It is worth noting that MUA-RS might fall slightly behind in specific metrics like Precision compared to Neural Collaborative Filtering and LightGCN, emphasizing that no single model excels across all dimensions. These results underscore the nuanced trade-offs inherent in recommendation systems, where each algorithm possesses unique strengths that contribute to an intricate landscape of personalized recommendations.

*5.2. Comparative Analysis*

The experimental results showcase the remarkable performance of MUA-RS across various evaluation metrics. With high Precision, Recall, NDCG, and competitive MAE and MSE scores, MUA-RS demonstrates its ability to deliver personalized recommendations that resonate with users' tastes. The integration of multimodal information enables the system to outperform traditional collaborative filtering methods. While UBCF provides reasonably accurate recommendations, it often suffers from the "cold-start" problem and sparsity issues, impacting its effectiveness for new users or items. Similar to UBCF, IBCF faces limitations with cold-start issues and sparse data, leading to potential inaccuracies in its recommendations. While effective in some cases, SVD-based approaches struggle with scalability and cold-start issues, hindering their performance on larger datasets. NCF exhibits competitive results, particularly concerning precision and recall. However, it may not fully exploit multimodal cues like MUA-RS. Finally, although LightGCN improves recommendations over traditional methods, it lacks the personalized adaptation and visual understanding found in MUA-RS. In summary, MUA-RS excels by integrating user ratings and movie poster images through user-adaptive filters, enhancing recommendation quality.

MUA-RS's innovation in leveraging multimodal cues allows it to rival even advanced approaches like NCF and LightGCN. This comparative analysis highlights MUA-RS's significance in advancing personalized recommendation systems with a multimodal and user-adaptive paradigm.

In the following scenarios, MUA-RS's ability to incorporate visual information, diversify recommendation lists, and adapt to user preferences through user-adaptive filters positions it as a powerful recommendation system that can surpass traditional methods in delivering personalized and engaging content to users:

- User Preferences Across Genres: Imagine a user who has consistently shown a preference for both action-packed thrillers and heartwarming romantic comedies. While traditional methods might struggle to strike a balance between these seemingly disparate genres, MUA-RS excels in capturing intricate preferences by utilizing visual attributes from movie posters. This enables MUA-RS to recommend action films with suspenseful visuals and romantic comedies with vibrant and cheerful poster designs, ensuring a well-rounded top-N list that resonates with the user's multifaceted tastes;
- Novelty and Exploration: Users often appreciate discovering new and diverse content. MUA-RS's ability to diversify recommendation lists by incorporating movie poster images can be particularly advantageous here. For example, if a user has expressed interest in crime dramas and documentary films, MUA-RS can identify visually captivating documentaries on crime-related subjects, providing a unique blend of genres that piques the user's curiosity and encourages exploration;
- Visual Appeal and Aesthetics: Some users are drawn to movies not only for their content but also for their visual aesthetics. MUA-RS shines in this regard by considering the visual attributes of movie posters. For instance, if a user has an affinity for movies with striking visual landscapes, MUA-RS can recommend films that boast captivating poster images depicting breathtaking scenery, enhancing the overall recommendation quality compared to methods solely reliant on text-based metadata;
- Adaptation to User Preferences: Consider a user who favors classic films from the 1950s and contemporary action blockbusters. Traditional methods may struggle to accommodate such diverse preferences. MUA-RS, with its user-adaptive filters, can accurately capture the user's distinct inclinations and recommend classic movies with vintage poster aesthetics alongside visually intense modern action films, catering to both aspects of the user's taste;
- Cultural Context and Themes: Users often have cultural or thematic preferences that might be missed by traditional methods. MUA-RS, with its visual understanding, can identify subtle cues in movie posters that align with specific cultural themes or nuanced storylines. For instance, if a user is a fan of international cinema and historical dramas, MUA-RS can identify visually captivating posters that convey these themes, enriching the recommendation list with culturally relevant content.

*5.3. Visual Influence Analysis*

The impact of visual stimuli is empirically evaluated through a rigorous analysis using the MovieLens dataset [23]. By focusing on movie posters as prominent visual cues, the study delves into the influence of these visuals on user preferences. The analysis involves examining neuron interactions within the CNN architecture to dissect the intricate mechanisms through which visual attributes shape and guide user choices.

In the domain of personalized movie recommendations, the use of specialized filters holds the potential to uncover latent user preferences by discerning intricate visual patterns. In this context, the User-Adaptive Filtering Module creates two distinct $5 \times 5$ filters. One filter is designed to detect darker and more intense aspects, while the other is crafted to capture brighter and more vibrant attributes. These filters play a fundamental role in capturing user-adaptive patterns that resonate with individual cinematic inclinations. The filter designed to detect dark and ominous lighting effectively accentuates shadowy nuances and ominous lighting within movie posters. This inference reflects a user's inclina-

tion towards suspenseful and intense genres. On the other hand, the bright and vibrant attributes detector filter focuses on amplifying vivid colors and luminosity, mirroring a user's proclivity for visually captivating and vibrant cinematic experiences. These two filters are visually represented in Figure 2.

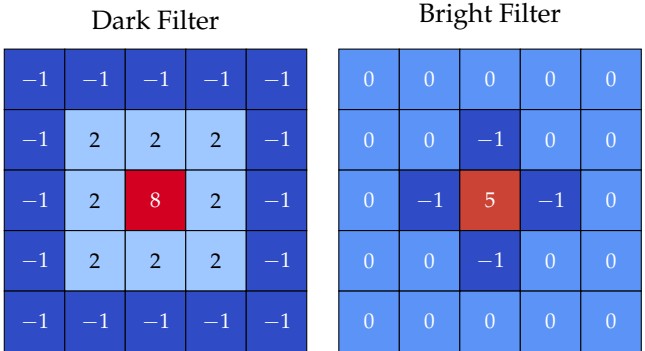

**Figure 2.** Two adaptive filters created by the model.

The notable advantage of these filters lies in their seamless integration within the MUA-RS architecture, which enhances the recommendation process with nuanced insights drawn from visual cues. As users interact with the system, their profiles are encoded to dynamically select and apply the appropriate filter, allowing the network to astutely identify cinematic attributes that align with individual preferences. This synergy empowers MUA-RS to effectively extract user-adaptive patterns from movie posters, resulting in a heightened sense of personalization in the recommendation generation process.

By skillfully deciphering the visual intricacies of movie posters, these filters bring to light the contours of user preferences that might otherwise remain latent. As a result, MUA-RS not only enhances recommendation accuracy but also forges a deeper connection between users and their cinematic choices. The incorporation of these filters underscores the potency of visual information in shaping personalized recommendations and sets the stage for a refined and holistic approach to the realm of movie recommendation systems.

In Figure 3, the application of the Dark and Bright Filters is depicted, tailored to detect distinct visual attributes within images. The aim is to demonstrate their effectiveness in capturing salient patterns and enhancing personalized recommendation systems. The Dark Filter, which has learned to discern darker and more intense aspects, is applied to an image featuring a dense and shadow-laden forest. Remarkably, the filter adeptly accentuates the intricate textures of tree trunks, effectively capturing the ominous lighting and shadow interplay inherent in such environments. In contrast, the Bright Filter, finely tuned to identify brighter and vibrant attributes, is deployed on an image depicting colorful balloons adorning a house against a clear blue sky, akin to the whimsical ambiance portrayed in the movie "Up". Impressively, this filter vividly amplifies the balloons' radiant hues, illuminating the vibrancy and vividness of the scene. The discernible success of these filters in extracting contextually relevant attributes underscores their potential as integral components within recommendation frameworks. They provide a lens through which personalized patterns can be deciphered from visual cues. This empirical exploration highlights the versatility of tailored filters in discerning diverse user preferences, fostering a deeper level of personalization and enriching the dynamics of contemporary recommendation systems.

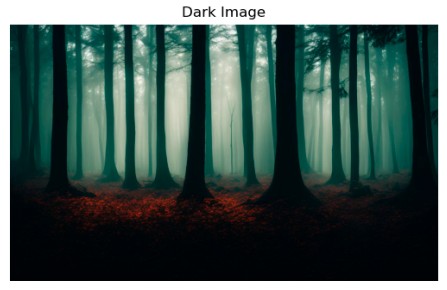
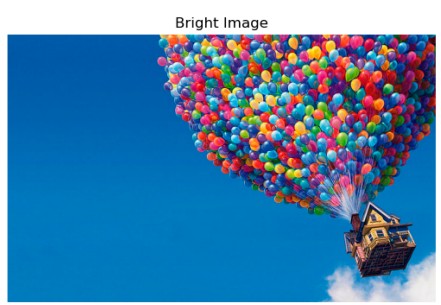

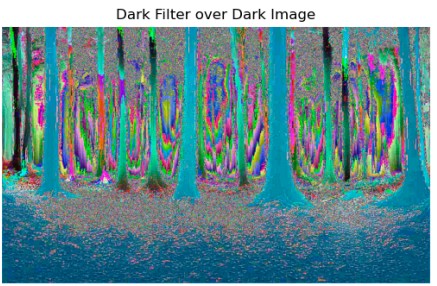
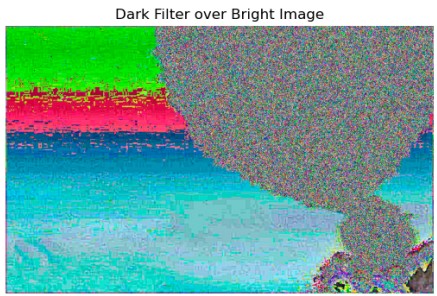

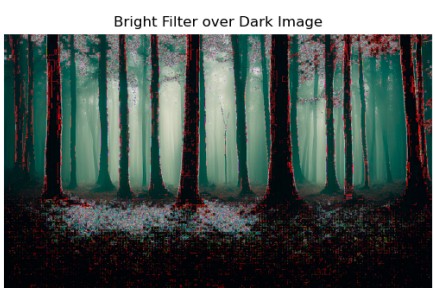
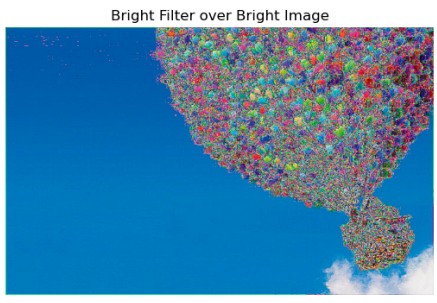

**Figure 3.** Dark and Bright Adaptive Filters applied over a dark and a bright image.

### 5.3.1. Personalized Visual Recommendation

The incorporation of the User-Adaptive Filtering Module within the MUA-RS recommendation architecture represents a significant advancement in the realm of personalized recommendations. Unlike conventional approaches that treat all users uniformly, this module empowers the neural network to dynamically generate a set of adaptive filters tailored to the unique preferences of each individual user. This adaptability enables MUA-RS to intricately adjust its feature extraction process to capture subtle nuances in user preferences, thereby enhancing the accuracy and relevance of recommendations.

Consider a scenario in the MovieLens 100K dataset where two users encounter the same movie poster. The User-Adaptive Filtering Module recognizes that the visual cues present in the poster may evoke distinct responses due to the users' differing tastes in movie genres. For instance, a poster highlighting action-packed scenes might entice a user who favors action movies, while the same poster could be less appealing to another user with a penchant for romantic comedies. By autonomously recognizing and adapting to such variations in user inclinations, the User-Adaptive Filtering Module not only reinforces the personalization of recommendations but also augments the system's capability to comprehend and cater to the intricate interplay of visual cues and user preferences. This dynamic adaptability encapsulates the essence of user-centric recommendation systems,

positioning MUA-RS at the forefront of precision and personalization in the realm of multimodal recommendation models.

To elucidate the advantages of the User-Adaptive Filtering Module, an analysis of two different user profiles learned by the model was carried out. This feature visualization allows us to discern the areas of greatest interest in the same image based on two different preferences: horror and Disney movies. For the analysis, the posters of two new test movies, which are not in the MovieLens 100K dataset, were taken into account: "Winnie the Pooh: Blood and Honey (2023)" (https://www.imdb.com/title/tt19623240 (accessed on 19 August 2023)) from the horror genre, and "Toy Story of Terror (TV Movie of 2013)", a Disney movie with a hint of kid-friendly horror (https://www.imdb.com/title/tt2446040 (accessed on 19 August 2023)).

The first poster for the film "Winnie the Pooh: Blood and Honey (2023)" portrays a figure standing against the backdrop of a nocturnal scene, illuminated by the headlights of a vehicle. The figure is clad in a black hoodie with distinctive bear ears, clutching a knife in one hand and holding a severed head in the other. The poster embodies a horror motif, with the silhouette of the individual foregrounded against a backdrop of a sinister forest. In contrast, the second poster showcases the central characters of the film "Toy Story of Terror (2013)", namely Woody, Buzz Lightyear, and Jessie. These characters are depicted in a state of profound trepidation before a vivid green light that illuminates their eyes. The backdrop of this poster captures a dark, nocturnal landscape featuring a full moon and an eerie castle.

The feature visualization showcased in Figure 4 provides insights into how the User-Adaptive Filtering Module tailors its feature extraction process based on user preferences. For the user with a penchant for horror movies, the visualization emphasizes eerie and intense aspects of the poster, aligning with the dark and ominous lighting detector filter learned by the system. In contrast, for the user inclined towards Disney movies, the visualization accentuates vibrant and cheerful attributes, aligning with the bright and vibrant attributes detector filter. This adaptive mechanism allows the system to amplify the visual cues that resonate with each user's preferences, thereby enhancing the personalization and relevance of recommendations.

In essence, the User-Adaptive Filtering Module brings a new dimension of personalization to the recommendation process, capitalizing on visual cues to tailor recommendations to individual tastes and preferences. This dynamic adaptation enriches the user experience by aligning the system's output with users' unique cinematic inclinations.

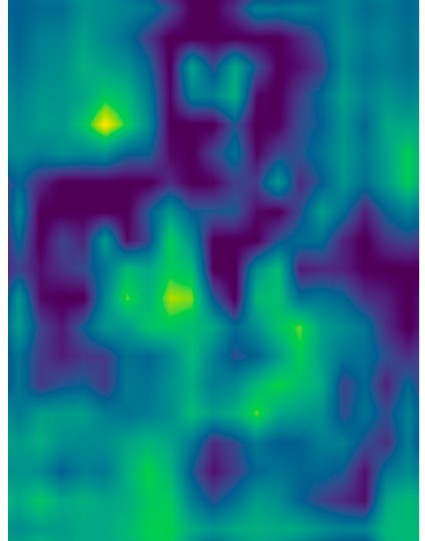 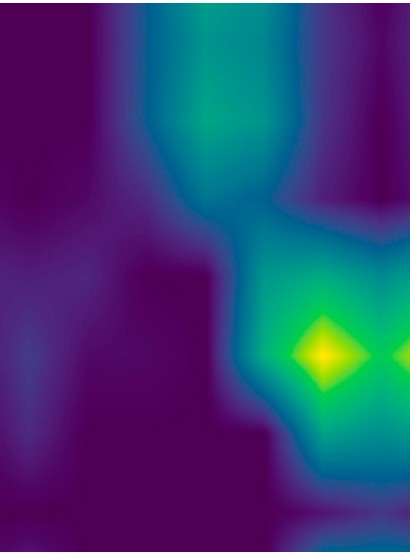

**Figure 4.** Activation maps for two different profiles users over "Toy Story of Terror (TV Movie of 2013)" poster.

The concept of multimodal neurons within the MUA-RS architecture exemplifies neural versatility at its finest. Trained on a diverse range of movie posters that encapsulate not only visual attributes but also textual and thematic elements, these neurons demonstrate a remarkable ability to transcend the boundaries of individual modalities. They respond not just to visual features, but also to the semantic cues that weave through images, text, and context.

MUA-RS effectively harnesses the power of these multimodal neurons to establish intricate associations, as evidenced by its capacity to extract the essence of "Toy Story (1995)" and seamlessly integrate it into the analysis of the "Toy Story of Terror! (2013)" movie poster. This ability highlights the neurons' agility in capturing inherent themes and characters that span across different iterations of a narrative. This synthesis speaks to the adaptability of the architecture, leveraging multimodal neurons to break free from singular data silos and uncover insights that span multiple dimensions.

The prominence of these multimodal neurons underscores their pivotal role in fostering a comprehensive understanding of the intricate interplay between user preferences and the multifaceted attributes of movies. By tapping into the insights encapsulated within these neurons, MUA-RS is empowered to provide more informed and personalized recommendations. This synthesis of diverse modalities enriches the recommendation process, allowing MUA-RS to provide recommendations that resonate deeply with users' preferences and inclinations.

5.3.2. Polysemantic Neurons

The emergence of "polysemantic neurons" within neural network architectures introduces a fascinating dimension to the interpretability and functionality of these systems. These neurons possess the unique ability to respond to a wide array of inputs that may seem unrelated on the surface. This multi-functionality of neurons adds complexity to neural network behavior and has significant implications, especially within architectures like MUA-RS (Multimodal User-Adaptive Recommender System) that leverage adaptive filters to enhance recommendations.

In the context of MUA-RS, polysemantic neurons play a crucial role in dynamically crafting User-Specific Adaptive Filters. These filters are designed to capture subtle nuances in user preferences and guide the recommendation process accordingly. For the horror movie enthusiast, MUA-RS's polysemantic neurons exhibit unique responsiveness. Some neurons might converge to detect and extract latent attributes associated with darkness, suspense, and horror elements. Others may respond to more subtle cues like ominous lighting, eerie atmospheres, or the presence of iconic horror motifs. The collective activity of these polysemantic neurons contributes to the generation of adaptive filters that emphasize the unique characteristics of horror movies, aligning perfectly with the user's preferences. In contrast, the polysemantic neurons for the Disney enthusiast take on a different role. They might converge to detect attributes like vibrancy, warmth, and enchantment—attributes closely tied to the Disney movie experience. These neurons could respond to vivid colors, whimsical imagery, and the presence of beloved Disney characters. The interplay of these polysemantic neurons leads to adaptive filters that highlight attributes intrinsic to Disney movies, aligning seamlessly with the user's inclinations.

The significance of polysemantic neurons within MUA-RS becomes evident in their adaptability to learn and encode user-specific preferences from seemingly unrelated inputs. This adaptability is a pivotal factor in generating refined recommendations that resonate with individual tastes. The polysemantic nature of these neurons allows MUA-RS to discern latent attributes across diverse genres, enabling it to capture the essence of horror and Disney movies alike. Ultimately, the advantage of polysemantic neurons in MUA-RS lies in their role as facilitators of personalization, ensuring that adaptive filters are finely tuned to each user's distinct cinematic inclinations. This capacity amplifies the architecture's effectiveness in providing recommendations that are both accurate and deeply aligned with individual preferences.

## 6. Discussion and Conclusions

In this study, a novel CNN-based neural network architecture called Multimodal User-Adaptive Recommender System (MUA-RS) was introduced to leverage multimodal data for enhanced recommendations. The empirical analysis on the MovieLens dataset highlighted the significant role of visual cues, particularly movie posters, in shaping user preferences and enriching the recommendation process.

The key advantages of MUA-RS include the following:

- User-Adaptive Filtering: The introduction of the User-Adaptive Filtering Module allows the model to dynamically adjust its feature extraction process for individual users. This personalized adaptation enhances the model's ability to capture nuanced preferences and improves the accuracy of recommendations;
- Enhanced Personalization: By integrating both explicit ratings and visual preferences, MUA-RS provides recommendations that are not only accurate but also closely aligned with each user's unique taste. This approach bridges the gap between implicit visual cues and explicit user ratings;
- Fine-grained Feature Learning: The user-adaptive filters enable MUA-RS to focus on specific visual attributes that resonate with users. This fine-grained feature learning contributes to a deeper understanding of item characteristics and user preferences;
- Improved Accuracy: The incorporation of user-adaptive filters enhances the accuracy of feature extraction, leading to more precise and relevant recommendations that capture the intricate interplay of visual cues and user preferences;
- Flexibility and Extensibility: MUA-RS's architecture is designed to accommodate various modalities beyond ratings and images, making it adaptable to diverse recommendation scenarios and potentially enriching the recommendation process further.

The introduction of multimodal neurons capable of responding to both textual and visual cues within the MUA-RS architecture represents a significant advancement. This capability enhances the model's understanding of movie attributes, themes, and implicit connections. Consequently, it results in more personalized and insightful recommendations that resonate deeply with users, bridging the gap between explicit preferences and implicit affinities.

Furthermore, MUA-RS's adaptability to user preferences through user-adaptive filters sets it apart from traditional recommendation methods. This dynamic approach tailors recommendations to individual users' unique inclinations, enhancing personalization and user engagement.

In conclusion, the Multimodal User-Adaptive Recommender System (MUA-RS) presents a pioneering approach to recommendation systems by seamlessly integrating explicit ratings, item images, and user-specific adaptive filters. This work contributes not only to the advancement of recommendation algorithms but also sheds light on the intricate interplay between visual cues and user preferences. Future research can extend this work to broader domains, explore more advanced techniques for multimodal integration, and delve deeper into the mechanisms behind user-adaptive filters for even more accurate and personalized recommendations. The MUA-RS architecture lays a solid foundation for the future of recommendation systems that take full advantage of multimodal data and user-centric adaptation.

**Funding:** The research was funded by Universidad Tecnica Federico Santa Maria under Project "Proyectos Internos USM 2023 PI_LII_23_01".

**Data Availability Statement:** Not applicable.

**Acknowledgments:** The author sincerely appreciates the support by Universidad Tecnica Federico Santa Maria under Project "Proyectos Internos USM 2023 PI_LII_23_01".

**Conflicts of Interest:** The authors declare no conflict of interest.

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
