# Peer review of "A Multimodal User-Adaptive Recommender System"

_electronics, doi:10.3390/electronics12173709_

Round 1

Reviewer 1 Report

This paper introduces an original Convolutional Neural Network (CNN) architecture that leverages multimodal information, connecting user ratings with product images to elevate item recommendations. A key innovation of this model is the User-Adaptive Filtering Module, a dynamic component that utilizes user profiles to generate personalized filters.

My Comments

1- Despite the article's single authorship, I noticed that most of the statements were in the plural. For instance, our model, we demonstrate, if we are recommending, we propose, We crawled, and so on 

2- The paper needs to be shortened and reorganized because of how poorly it is arranged and presented.

3- Extensive editing of English language required

4-The author does not explain where these data originated from or what the formulas for precision, recall, MSE, and other metrics are in Table 2, which shows a comparative review of recommendation algorithms. How can readers believe these values?

5- The author has to explain how this architecture addresses issues like overcoming the "cold-start problem," increasing engagement, dealing with data and storage challenges, and others.

6- Despite their potential, there are also challenges in developing multimodal user-adaptive recommender systems. Integration and fusion of various modalities, handling scale and diversity of data, and ensuring privacy and security of user information are areas that require attention and research. The subject matter needs to be covered in depth by the author.

Extensive editing of English language required

Author Response

Response to Reviewer 1 Comments

Dear Reviewer,
Your encouraging, critical, and constructive feedback on the manuscript is truly appreciated. The insights provided have been of immense value in refining the content and presentation of the work.

Point 1: Despite the article's single authorship, I noticed that most of the statements were in the plural. For instance, our model, we demonstrate, if we are recommending, we propose, We crawled, and so on.

Response 1: Thank you for noticing it. As far as I know, in academic writing, especially in computer science, is common and acceptable to use the plural form in scientific papers even if you are the sole author. However, to adhere to the stylistic conventions of the journal, the manuscript has been meticulously revised to utilize the passive voice throughout.

Point 2: The paper needs to be shortened and reorganized because of how poorly it is arranged and presented.

Response 2: Your feedback is immensely valuable. The manuscript underwent a comprehensive revision, entailing the removal of certain paragraphs and the reorganization of sections, resulting in a more coherent and succinct presentation.

Point 3: Extensive editing of English language required.

Response 3: Your point is well taken. A thorough revision of the English language was undertaken to ensure the text's clarity and coherence.

Point 4: The author does not explain where these data originated from or what the formulas for precision, recall, MSE, and other metrics are in Table 2, which shows a comparative review of recommendation algorithms. How can readers believe these values?

Response 4: Thanks for raising valid concerns about transparency. Section 4.3, titled "Evaluation Methodology," has been meticulously revised to provide a comprehensive explanation of the data's origin and the formulas for the evaluation metrics in Table 2. Moreover, section 4.4 delves into a detailed elucidation of the evaluation metrics employed.

Point 5: The author has to explain how this architecture addresses issues like overcoming the "cold-start problem," increasing engagement, dealing with data and storage challenges, and others.

Response 5: Thank you very much for this insightful comment, as it significantly enriches the manuscript's scientific depth. In response, a new paragraph has been incorporated into section 3.2, where the architecture's approach to addressing challenges such as the "cold-start problem," engagement enhancement, and data and storage challenges is thoroughly explained. This addition aims to provide a comprehensive understanding of the architecture's capabilities in tackling these critical issues.

Point 6: Despite their potential, there are also challenges in developing multimodal user-adaptive recommender systems. Integration and fusion of various modalities, handling scale and diversity of data, and ensuring privacy and security of user information are areas that require attention and research. The subject matter needs to be covered in depth by the author.

Response 6: Very thanks for acknowledging the potential of the work. The challenges that have been highlighted are indeed significant facets within the domain of multimodal user-adaptive recommender systems. Given the innovative nature of the proposed architecture, a commitment has been made to explore these challenges in future endeavors, thereby fostering a more comprehensive grasp of their implications and potential solutions. The valuable insight provided by you has genuinely enriched the perspective and will serve as a guiding force for the research directions moving forward.

Reviewer 2 Report

In this paper, the author proposes a content-based recommender system based on the newly proposed Convolutional Neural Network (CNN) architecture that uses multimodal information for accurate recommendations within the recommender system. This architecture combines user reviews with product images to improve future item recommendations. The paper is interesting, but has some limitations that should be addressed. I don't know if this is a major or minor revision, but I don't think it should be a problem to make the necessary changes.

1.

The literature review is a bit outdated. Expand it with more recent scientific papers from recent years.

2.

The proposed system is succinctly described. It would be interesting to add an example of how a user is validated within the MovieLens100K dataset and which movies are really shown to them as the ones they should be interested in. Describe the whole process (section 3) in a bit more detail with an example. Especially training process with some example.

3.

Section 5. table 2.

Is MUA-RS compared with the other algorithms listed in Table 2 as part of your system, or are they separate algorithms that have already been evaluated and whose results are merely listed there?

Why is this important? When your entire system, which has MUA-RS, is compared to other systems that have been tested with the MovieLens100K dataset, you have an advantage because you have downloaded the cover images for all the movies and have more information than other algorithms. It is very important to clarify this in the paper. How was the evaluation done? Did you simply download the results?

To do:

It would also be interesting to see how your system stands up in the results if you did not include images in your system and thus only have the MovieLens100K dataset available. Please expand the test and evaluation to include another row in Table 2 where you would test your system without using cover images of movies.

I do not want to say that this is not good, on the contrary... this would additionally show to readers the difference in the quality of the system without images (MovieLens100K dataset only) and with images.

4.

Figure 4. and the explanation of how your system detects similarities in the same movie genre.

In my humble opinion, it would be better to replace the example of horror movies with a classic genre such as comedy, action, or something similar that perhaps a larger number of viewers (readers of this paper) can identify with.

Author Response

Response to Reviewer 2 Comments

Dear Reviewer,

Your invaluable feedback, which has played a pivotal role in refining the manuscript and elevating the scholarly contribution of the work, is sincerely appreciated and will undoubtedly shape the continued development of this research.

Point 1: The literature review is a bit outdated. Expand it with more recent scientific papers from recent years.

Response 1: Your suggestion is greatly appreciated, and it has been duly considered. The literature review has been updated to include recent publications that are relevant to the state-of-the-art analysis.

Point 2: The proposed system is succinctly described. It would be interesting to add an example of how a user is validated within the MovieLens100K dataset and which movies are really shown to them as the ones they should be interested in. Describe the whole process (section 3) in a bit more detail with an example. Especially training process with some example.

Response 2: Your insightful comment on providing a detailed example for the validation process within the MovieLens100K dataset, along with an elaboration of the training process, has been immensely beneficial. In response, a new paragraph has been thoughtfully incorporated into section 3. This addition expounds upon the training process, accompanied by a pertinent user example, thereby enhancing the comprehensibility and clarity of the methodology.

Point 3: Section 5. table 2.

Is MUA-RS compared with the other algorithms listed in Table 2 as part of your system, or are they separate algorithms that have already been evaluated and whose results are merely listed there?

Why is this important? When your entire system, which has MUA-RS, is compared to other systems that have been tested with the MovieLens100K dataset, you have an advantage because you have downloaded the cover images for all the movies and have more information than other algorithms. It is very important to clarify this in the paper. How was the evaluation done? Did you simply download the results?

To do:

It would also be interesting to see how your system stands up in the results if you did not include images in your system and thus only have the MovieLens100K dataset available. Please expand the test and evaluation to include another row in Table 2 where you would test your system without using cover images of movies.

I do not want to say that this is not good, on the contrary... this would additionally show to readers the difference in the quality of the system without images (MovieLens100K dataset only) and with images.

Response 3: Thanks for raising valid concerns about transparency. Section 4.3, titled "Evaluation Methodology," has been meticulously revised to provide a comprehensive explanation of the data's origin and the formulas for the evaluation metrics in Table 2. To clarify this point, in order to maintain consistency across the algorithms being compared, a deliberate effort has been made to implement and execute all methods within a virtual environment built on the Python 3 kernel. Moreover, to address your insightful query, the proposed model's performance without images was also tested and its results were thoughtfully integrated into Table 2. Your suggestion has undeniably contributed to a more comprehensive analysis of the proposed approach.

Point 4: Figure 4. and the explanation of how your system detects similarities in the same movie genre.

In my humble opinion, it would be better to replace the example of horror movies with a classic genre such as comedy, action, or something similar that perhaps a larger number of viewers (readers of this paper) can identify with.

Response 4: Thank you for the suggestion, is a very good advice and was considered. Recognizing the value of your suggestion, the example added in section 3 is based on classic genres such as comedy and action, which resonate more widely with readers. This adjustment aligns the manuscript with a broader reader base, ensuring enhanced clarity and understanding.

Round 2

Reviewer 1 Report

The submitted scientific paper meets the standards and criteria necessary for publication.

Minor editing of English language required

Reviewer 2 Report

The paper is accepted for publication in this form